# Activation of Small Conductance Ca^2+^-Activated K^+^ Channels Suppresses Electrical and Calcium Alternans in Atrial Myocytes

**DOI:** 10.3390/ijms26083597

**Published:** 2025-04-11

**Authors:** Giedrius Kanaporis, Lothar A. Blatter

**Affiliations:** Department of Physiology and Biophysics, Rush University Medical Center, 1750 W. Harrison Street, Chicago, IL 60612, USA; lothar_blatter@rush.edu

**Keywords:** atria, alternans, action potential, calcium, small conductance Ca^2+^-activated K^+^ channels, long QT syndrome

## Abstract

Small conductance Ca^2+^-activated K^+^ (SK) channels are expressed in atria and ventricles. However, the data on the contribution of SK channels to atrial action potential (AP) repolarization are inconsistent. We investigated the effect of SK channel modulators on AP morphology in rabbit atrial myocytes and tested the hypothesis that pharmacological activation of SK channels suppresses pacing-induced Ca^2+^ transient (CaT) and AP duration (APD) alternans. At the cellular level, alternans are observed as beat-to-beat alternations in contraction, APD, and CaT amplitude, representing a risk factor for arrhythmias, including atrial fibrillation. Our results show that SK channel inhibition by apamin did not affect atrial APD under basal conditions. However, SK channel activation by NS309 significantly shortened APD, indicating the expression of functional SK channels. Moreover, the activation of SK channels reduced CaT amplitude and sarcoplasmic reticulum Ca^2+^ load. Activation of SK channels also suppressed pacing-induced CaT and APD alternans. K_V_7.1 potassium channel inhibition, simulating long QT syndrome type-1 conditions, increased the risk of atrial CaT alternans, which was abolished by the activation of SK channels. In summary, our data suggest that pharmacological modulation of SK channels can potentially reduce atrial arrhythmia risk arising from pathological APD prolongation.

## 1. Introduction

Small conductance Ca^2+^-activated K^+^ (SK) channels are expressed in both atria and ventricles [1,2,3,4,5,6,7,8], including in humans [1,3]. The family of SK channels consists of three isoforms (SK1,2,3), which are characterized by low conductance around 10 pS and gated by intracellular Ca^2+^. In theory, Ca^2+^-dependent gating of SK channels can directly link the regulation of intracellular Ca^2+^ signaling and cell membrane potential and play an important role in Ca^2+^ homeostasis through membrane potential-dependent feedback mechanisms. However, current data on the role of SK channels in atria are inconsistent, as some studies have demonstrated that SK channels can contribute to atrial action potential (AP) repolarization [1,2,3,4,5,9,10,11], while other studies [12,13,14] suggest that under basal conditions SK channels do not contribute to atrial AP morphology (for recent reviews of the role of SK channel in the heart and cardiac disease see, e.g., [15,16]). While these divergent results might reflect differences in animal species and experimental conditions, it remains a fact that the physiological and pathophysiological roles of SK channels in the heart are poorly understood. Therefore, we set out to investigate SK channels’ presence and functional role in mammalian (rabbit) atrial cells using pharmacological tools, electrophysiological (patch clamp), and Ca^2+^ imaging methods. Specifically, we focused on SK channels’ role in AP morphology under basal conditions and after pharmacological manipulation of AP duration (APD). We also investigated how SK channel activity feedbacks on cytosolic Ca^2+^ transients (CaT).

SK channels have recently gained increased attention for their involvement in cardiac arrhythmia, including atrial fibrillation (AF), and as potential antiarrhythmic therapeutic targets [17,18,19,20,21]. Cardiac arrhythmia, and specifically AF, are causally linked to cardiac alternans. Cardiac alternans are observed as beat-to-beat alternation in contraction strength (mechanical alternans), CaT amplitude (CaT alternans), and action potential duration (APD or electrical alternans) at constant beating frequency. Numerous experimental [22,23] and clinical studies [24,25,26,27] have provided convincing evidence that AP alternans in atria precede AF episodes, leading directly to AF or facilitating the transition from atrial flutter to AF. It is well established that both arrhythmogenic triggers and an appropriate substrate are required for the initiation and perpetuation of AF [28,29]. Atrial alternans play a major role in generating proarrhythmic substrate and facilitating re-entry phenomena [25,26,28,30]. In addition, there is emerging evidence for a significant role of SK channels in AF. For example, Ellinor et al. [31] suggested that mutation in SK3 channels can be directly linked to familial AF. An increase in SK current was reported in atrial myocytes obtained from patients with AF [32], and this phenotype was recapitulated in canine [14] and porcine [11] rapid-pacing AF models. However, other studies have found decreased expression of SK channels in patients with chronic atrial fibrillation [33,34]. Furthermore, overexpression [4] and knockout [35] of SK channels predisposed mice to AF, likely acting through different mechanisms. It was hypothesized that a decrease in SK current could trigger AF by prolonging APD, thereby increasing the risk of delayed afterdepolarizations. Conversely, overexpression was suggested to shorten APD and effective refractory period, generating an arrhythmogenic substrate for AF. Here, we tested the hypothesis of whether activation of SK channels abolishes or reduces the risk of pacing-induced APD and CaT alternans. We also explored the perspective of pharmacological SK channel manipulation as a therapeutic tool for proarrhythmic alternans prevention.

Long QT (LQT) syndrome entails a group of cardiac pathologies with the common feature of a prolonged APD caused by a variety of ion channel defects (for review, see [36,37]). In contrast to the ventricle, the pathophysiology of LQT syndrome in the atria has been insufficiently investigated. Polymorphic atrial tachycardia and AF are commonly observed in LQT syndrome patients [38,39,40,41,42,43,44,45,46,47,48,49], which can further aggravate the risk of clinical complications and lead to even higher morbidity and mortality. We, therefore, tested the hypothesis of whether in a model of pharmacologically induced LQT syndrome type-1 activation of SK channels can protect from the risk of proarrhythmic atrial alternans.

In summary, the goals of our study were several folds: (1) demonstrate functional SK channels in rabbit atrial myocytes that can be pharmacologically activated and explore the interplay between CaT and AP regulation mediated by SK channels; (2) investigate the effect of SK channel activation as a protective measure against proarrhythmic APD and CaT alternans; and (3) test the potential of SK channel activation to reduce arrhythmia risk as a potential target for LQT syndrome therapy.

## 2. Results

### 2.1. Effects of SK Channel Inhibition and Activation on Atrial AP Repolarization

The physiological role of SK channels in the heart, including the atria, is poorly understood. Several studies have demonstrated that SK channels contribute to atrial repolarization [1,3,9,10,11], while other investigations have found little or no change in AP morphology upon SK channel inhibition [12,13,14]. To determine the contribution of SK channels to rabbit atrial repolarization, APs were measured from current-clamped atrial myocytes paced at 0.5 Hz in control and in the presence of SK channel blocker apamin (100 nM; Figure 1(Aa)). APD was quantified at 70% and 90% repolarization levels (Figure 1(Ab)). Apamin had no effect on atrial APD (Figure 1A, n/N = 15/10), resting membrane potential (V_m_) (−88.6 ± 3.1 mV in control; −88.8 ± 2.6 mV in apamin; *p* = 0.6092; paired *t*-test) or AP amplitude (135.9 ± 4.4 mV in control; 136.3 ± 4.1 mV in apamin; *p* = 0.4661).

Next, we tested if the pharmacological activation of SK channels affects atrial APD (Figure 1B). Application of SK channel activator NS309 (2 µM) resulted in a pronounced APD shortening. Activation of SK channels shortened APD_70_ from 359 ± 42 ms in control to 261 ± 22 ms (n/N = 6/2, *p* = 0.0007; paired *t*-test), while APD_90_ shortened from 500 ± 63 ms to 389 ± 73 ms (*p* = 0.0010; Figure 1(Bb)). NS309 had no effect on resting V_m_ (−91.4 ± 1.8 mV in control; −90.9 ± 1.6 mV in NS309; n/N = 6/2, *p* = 0.1859; paired *t*-test) or AP amplitude (138.1 ± 1.5 mV in control; 137.5 ± 2.0 mV in NS309; *p* = 0.2789). These results indicate that, while under basal conditions SK channels have only a minor effect on atrial repolarization, functional SK channels can be activated pharmacologically and are expressed in atrial cells at a density sufficient to cause significant APD shortening.

### 2.2. Activation of SK Channels Suppresses CaT and APD Alternans

Alternans is a well-established risk factor for cardiac arrhythmias, including AF [23,24,25,26]. AP morphology was shown to modulate the risk of cardiac alternans [50,51]. Therefore, we tested if SK channel activation in atrial myocytes has an effect on APD and CaT alternans. APD and CaT alternans were induced by increasing the pacing frequency. Figure 2A shows CaT traces recorded from the same field-stimulated atrial cell in control, in the presence of the SK channel activator NS309 (2 µM) and after the washout of NS309. SK channel activation consistently abolished or attenuated the degree of atrial CaT alternans. NS309 reduced mean CaT alternans ratio (AR) from 0.37 ± 0.16 to 0.14 ± 0.13 (Figure 2B; n/N = 17/6; *p* < 0.0001; Tukey’s mixed-effects multigroup comparison). In a subset of cells (n/N = 9/3) a washout of NS309 resulted in an increase in CaT AR to 0.30 ± 0.32, which was statistically not different from control AR.

Activation of SK channels by NS309 also abolished or reduced the degree of pacing-induced APD alternans. Figure 3A shows APs recorded from a current-clamped myocyte in control and after application of NS309. To quantify AP alternans, we calculated APD_70_ and APD_90_ ratios by dividing the longer APD by the shorter APD of alternating AP pairs (Figure 3B). Accordingly, ratio values of 1 indicate the absence of alternans. In the presence of NS309, the APD_70_ alternans ratio decreased from 1.28 ± 0.1 to 1.05 ± 0.04 (*p* = 0.0084; paired *t*-test), and the APD_90_ ratio decreased from 1.27 ± 0.12 to 1.04 ± 0.05 (*p* = 0.0104).

Our data clearly demonstrate that APD shortening due to SK channel activation reduced or completely abolished pacing-induced atrial CaT and APD alternans.

### 2.3. Effects of NS309 on CaT Alternans and APD Are Induced by SK Channel Activation

The specificity of pharmacological agents used in research is a common and valid concern. To determine whether the effects of NS309 were indeed elicited selectively by the activation of SK channels, we performed experiments where the SK channel blocker apamin was applied together with NS309. Figure 4(Aa) shows AP traces in control, in apamin (100 nM), and in the presence of apamin together with NS309 (2 µM). The data show that the presence of apamin prevented NS309 from shortening the AP. In addition, Figure 4(Ab) shows that apamin largely reversed the APD shortening (APD_70_ and APD_90_) elicited with NS309. We also tested if SK channel inhibition with apamin prevents the rescuing effect of NS309 on CaT alternans (Figure 4B). Once stable pacing-induced alternans were established, CaT alternans were recorded for at least 1 min under control conditions, and then myocytes were superfused with apamin for 3–4 min, followed by simultaneous exposure to apamin and NS309. After 3–4 min, apamin was removed, leaving the same cells exposed to NS309 alone. During these experiments, mean CaT AR in control was 0.43 ± 0.19, 0.42 ± 0.19 in apamin, and 0.34 ± 0.13 in the presence of apamin together with NS309 (Figure 4(Bb); n/N = 9/5, no statistically significant differences between CaT ARs for these different experimental conditions; Tukey’s multiple groups comparison test). When cells were subsequently exposed to NS309 alone, the mean CaT AR decreased to 0.12 ± 0.09 (*p* = 0.0023 vs. control and *p* = 0.0037 vs. NS309 + apamin), similar to the above reported (Figure 2) decrease in AR. From the observation that the effects of NS309 were abolished by SK channel inhibition, we conclude that NS309 induced APD shortening and rescue of CaT alternans are caused by the activation of SK channels.

### 2.4. Effect of SK Channel Activation on CaT Properties

We tested if activation of SK channels affects intracellular Ca^2+^ release in atrial myocytes (Figure 5). CaTs (Figure 5A) were monitored in field-stimulated cells paced at 0.5 Hz (i.e., at a frequency that did not elicit alternans). In two sets of experiments exposure to NS309 (2 µM) decreased CaT amplitude (ΔF/F_0_) from 3.22 ± 0.87 to 2.78 ± 0.86 (Figure 5(Ba); n/N = 11/3; *p* < 0.0006; Tukey’s multiple groups comparison test) and from 3.72 ± 1.10 to 2.96 ± 0.77 (Figure 5(Bb); n/N = 9/3; *p* < 0.0048). In addition, the effect of NS309 on CaT amplitude was abolished by the subsequent application of two different SK channel blockers: apamin (Figure 5(Ba); n/N = 11/3; *p* = 0.0065; Tukey’s multiple groups comparison test) or UCL1684 (Figure 5(Bb); n/N = 9/3, *p* = 0.0076) restored CaT amplitude to control levels, even in the maintained presence of NS309. The inhibition of SK channels alone (apamin 100 nM) had no effect on CaT amplitude (Figure 5C), and this is consistent with our findings that blocking SK channels does not affect AP repolarization (Figure 1A). In addition, the observation that the NS309 effect on CaT can be reversed by two different SK channel blockers provides further evidence that the effects of NS309 are elicited through SK channel activation.

Next, we tested whether the reduction in CaT amplitude was the result of decreased sarcoplasmic reticulum (SR) Ca^2+^ load. To evaluate [Ca^2+^]_SR,_ we measured the amplitudes of caffeine (10 mM) induced CaTs. Caffeine-induced CaTs were monitored in a paired manner in control and in NS309. To address a potential concern that a spontaneous decrease in [Ca^2+^]_i_ over time could affect the interpretation of our results, in half of the cells tested (four cells out of eight), caffeine-induced CaTs were first measured in control and subsequently in NS309, and in the other half of cells caffeine-induced CaTs were first elicited in NS309 treated cells and then recorded after washout of NS309. Data show (Figure 5D) that in control caffeine-induced CaT amplitude (ΔF/F_0_) was bigger (7.3 ± 1.6) than in NS309 (6.4 ± 1.4; n/N = 8/3, *p* = 0.0062, paired *t*-test) demonstrating that SK channel activation leads to a decrease in SR Ca^2+^ content.

### 2.5. APD Prolongation Increased the Risk for CaT Alternans, Which Can Be Attenuated by SK Channel Activation

Polymorphic atrial tachycardia and AF are commonly observed in LQT syndrome patients [38,39,40,41,42,43,44,45,46,47,48,49,52] and aggravate the risk of clinical complications. To mimic LQT type-1 conditions, which are linked to an increased risk of atrial arrhythmias in humans [48,52], we blocked Kv7.1 potassium channels with HMR1556 (1 µM). As expected, the application of HMR1556 resulted in the prolongation of the atrial AP (Figure 6(Aa)). APD_70_ prolonged from 303 ± 107 ms to 343 ± 98 ms (n/N = 8/5; *p* = 0.0233, paired *t* test) and APD_90_ from 433 ± 101 ms to 463 ± 109 ms (*p* = 0.0480) (Figure 6(Ab)). To test if SK channel activation could reduce the alternans risk in the LQT setting, field stimulation frequency was gradually increased until atrial myocytes exhibited stable CaT alternans, and then atrial myocytes were exposed to HMR1556 (1 µM) and to HMR1556 together with NS309. Figure 6(Ba) shows CaT alternans traces recorded in control, in HMR1556, and in HMR1556 together with NS309. The mean CaT AR in control was 0.40 ± 0.20. HMR1556 increased CaT AR to 0.50 ± 0.25 (n/N = 14/8; *p* = 0.0392; Tukey’s mixed effects multiple comparison test; Figure 6(Bb)). Subsequent application of NS309 during K_V_7.1 inhibition suppressed CaT alternans and reduced AR to 0.24 ± 0.24 (n/N = 13/7; *p* = 0.0079, Tukey’s mixed-effects multiple group comparison test). Consequently, our data demonstrate that simulated LQT syndrome type-1 in atrial myocytes increases the risk of proarrhythmic alternans, which can be prevented by the activation of SK channels.

## 3. Discussion

### 3.1. SK Channel Activity in Atrial Myocytes

SK channels are abundant in the central nervous system but are also present in the heart. Three subtypes of SK channels (genes *KCNN1*, *KCNN2*, and *KCNN3*) were found expressed in both atria and ventricles from various species [1,2,3,4,5], including rabbits [6,7,8]. Several studies in humans [1,3], dogs [9], pigs [10,11], and horses [5] have demonstrated that SK channels contribute to atrial AP repolarization. However, numerous studies found no or just a small effect of SK channel blockers on ventricular repolarization in healthy hearts [2,5,7,35,53,54]. The prevalent view is that in the ventricle, the role of SK channels becomes more prominent in pathologies associated with elevated [Ca^2+^]_i_, such as heart failure [55,56]. Consequently, it was suggested that SK channels could serve as a therapeutical target to modulate electric activity, specifically in atria, with little effect on ventricular electrophysiology [3,16,35]. However, the data on the physiological role of SK channels in atria are inconsistent and vary among studies. Our results indicate that under basal conditions in rabbit hearts, the inhibition of SK channels has no effect on APD duration in atrial myocytes (Figure 1). This result is in line with a study by Giommi et al. [12] that also demonstrated the absence of SK channel activation in rabbit atria. Moreover, no effect of SK channel inhibition on atrial APD was previously reported in humans [12], dogs [13,14], and rats [13]. The reason for such discrepancies between studies remains unclear, but they are likely to arise from species differences, varying experimental conditions, use of differing pharmacological agents, and variability of cardiac preparations. Furthermore, it was suggested that female hearts have a higher activity of SK channels, with differences becoming more noticeable during β-adrenergic stimulation [7,57].

Recently, we have demonstrated in rabbit ventricular myocytes that while inhibition of SK channels has no effect on ventricular APD, pharmacological activation of SK channels leads to significant APD shortening [54]. Here, we report that SK channel agonist NS309 has very similar effects in the atria. These results clearly indicate that fully functional SK channels are well expressed in both atrial and ventricular myocytes, even though the majority of these channels are not activated during the physiological excitation–contraction coupling cycle.

Activation of SK channels also had effects on intracellular Ca^2+^ cycling, leading to a decrease in CaT amplitude and SR Ca^2+^ load (Figure 5). We suggest that the effects of NS309 on Ca^2+^ handling are a consequence of APD shortening. It was demonstrated that SR Ca^2+^ content positively correlates with APD, and AP prolongation results in higher SR Ca^2+^ release due to enhanced Ca^2+^ entry through L-type Ca^2+^ channels [58] or through reverse mode Na^+^/Ca^2+^ exchange [59]. This is consistent with our previous study, using short and wide AP waveforms as voltage commands in voltage-clamped myocytes, demonstrating that cells paced with shorter APs had lower [Ca^2+^]_SR_ and reduced CaT amplitude [50].

To confirm the specificity of SK channel agonist NS309 action, we performed experiments applying NS309 together with two selective SK channel blockers: apamin and UCL1684. Pretreatment with SK channel blockers almost completely prevented the effect of NS309 on APD (Figure 4A) and on CaT alternans (Figure 4B). Also, the application of the SK channel blockers reversed the effects of NS309 on CaT amplitude (Figure 5B). These results are consistent with observations in our recent study in ventricular myocytes where, regardless of the sequence of drug application, inhibition of SK channels also counteracted the effects of NS309 on APD, CaT amplitude, and CaT and APD alternans [54]. Thus, both our studies confirm that NS309 indeed acts through SK channel activation. However, in both studies, we did not further explore inhibition or activation effects on specific subtypes of SK channels since currently available pharmacological modulators of SK channels have limited selectivity towards different subtypes. Apamin at a concentration of 100 nM ensures a near full block of all three SK channel subtypes, and apamin is highly selective for SK channels at this concentration [60]. It was suggested that heteromeric SK channels composed of SK2 and SK3 channel subunits in mice atria might be less sensitive to apamin but remain highly sensitive to UCL1684 [61]. However, in our studies, apamin and UCL1684 counteracted NS309-induced cellular effects to the same extent in both rabbit atrial (Figure 5B) and ventricular myocytes [54]. In summary, our observations in this and our previous study [54] suggest that, while SK channels do not contribute to AP repolarization under basal conditions, these channels are expressed in both atrial and ventricular myocytes and can be pharmacologically activated in both cardiac chambers. These results, taken together with the ambiguous reports on SK channel contribution to atrial AP morphology, suggest that the notion that SK channels might serve as atria-specific targets for AF prevention needs to be taken cautiously and requires further examination.

### 3.2. Cardiac Alternans, SK Channels and Atrial Arrhythmias

Cardiac alternans are characterized by repolarization and Ca^2+^ signaling irregularities throughout the myocardium and are closely linked to the development of cardiac arrhythmias. Clinical and experimental studies have provided strong evidence linking atrial alternans to AF [22,23,24,25,26,27], the most common cardiac arrhythmia that is affecting 1–2% of the US population, and AF prevalence is expected to increase dramatically as the population ages [62]. There are numerous lines of evidence for the role of SK channels in AF: atrial myocytes obtained from patients with AF revealed increased SK channel currents [32], a phenotype also found in experimental canine [14] and porcine [11] rapid-pacing AF models. This contrasts the observation that patients with chronic AF appear to have a decreased expression of SK channels [33,34]. Alternans contribute to the arrhythmogenic substrate required for sustained AF, and patients with AF are known to have a lower threshold for alternans well before the manifestation of arrhythmias [24,26,30]. Therefore, cardiac alternans prevention could serve as a valuable early therapeutic intervention with the potential to slow down or even prevent the development of cardiac diseases.

Here we demonstrate that the activation of SK channels in the atria abolished or attenuated CaT (Figure 2) and APD (Figure 3) alternans. It is well established that APD and CaT alternans typically coincide in time and space [63,64] (although exceptions have been demonstrated [65]). AP duration and shape control voltage-dependent Ca^2+^ handling mechanisms, such as Ca^2+^ entry through L-type Ca^2+^ channels and Ca^2+^ removal from the cytosol by Na^+^/Ca^2+^ exchange, thus affecting intracellular Ca^2+^ cycling. Intracellular Ca^2+^ dynamics, in turn, modulate membrane potential through Ca^2+^-dependent ion currents and transporters. In theory, due to the Ca^2+^ dependence of SK channel activation, these channels could also play a role in linking CaT and APD alternans. While SK channel inhibition failed to cause APD prolongation, we hypothesize that the suppression of alternans observed during SK channel activation is caused by APD shortening. This is consistent with our previous studies, demonstrating that prolongation of APD increased the risk for atrial alternans by modulating L-type Ca^2+^ channel activation kinetics and SR Ca^2+^ load [50] and that alternans can be attenuated by pharmacologically induced APD shortening via activation of K_V_7.1 or K_V_11.1 potassium channels [51]. The links between APD and alternans are complex and involve several potential mechanisms. We suggest that APD shortening leads to a longer diastolic interval, providing more time for SERCA to refill the SR with Ca^2+^ and for SR Ca^2+^ release to recover from refractoriness [66]. In addition, a longer diastolic interval also allows for full recovery of L-type Ca^2+^ current from inactivation and ensures consistent beat-to-beat Ca^2+^ entry and trigger for Ca^2+^-induced Ca^2+^ release. Further support for the hypothesis that alternans are suppressed due to APD shortening arises from our study on ventricular myocytes, where SK channel activation abolished CaT and APD alternans in field-stimulated and current-clamped atrial myocytes but had no effect on alternans in AP voltage-clamped ventricular cells [54].

### 3.3. Atrial Alternans in LQT Syndrome

SK channel activation-induced shortening of atrial APs may have various and complex consequences for atrial physiology and arrhythmia risk. It is known that excessive shortening of atrial APs may lead to AF by prolonging the diastolic interval, shortening the effective refractory period, and thus potentially increasing the risk of premature beats [25,26,28,30,67]. On the other hand, AP shortening has the potential to normalize AP morphology under pathological conditions where APD is abnormally long. APD prolongation is one of the most frequent cardiac electrical disorders, and multiple studies have demonstrated a positive correlation between prolongation of the QT interval and atrial arrhythmias [38,41,68,69]. Currently, mutations in 17 different genes have been demonstrated to be linked to congenital LQT syndrome [36,42]. In addition, APD prolongation is also caused by the adverse effects of a wide range of medications or by cardiac pathologies [70]. In contrast to the ventricle, the pathophysiology of LQT syndrome in atria has been barely investigated despite the fact that the risk of polymorphic atrial tachycardia and AF is enhanced in LQT syndrome patients [38,39,40,41,42,43,44,45,46,47,48,49] and animal models [71,72]. To test the hypothesis that the activation of SK channels can normalize AP morphology and reduce the degree of alternans under LQT syndrome conditions, we blocked delayed rectifier K_V_7.1 potassium channels (Figure 6). Loss of function mutations of K_V_7.1 channels results in LQT syndrome type-1, accounting for 40–50% of all congenital LQT syndrome cases [73]. Blocking K_V_7.1 channels resulted in APD prolongation (Figure 6A) and increased susceptibility to CaT alternans (Figure 6B), which was abolished or reduced by SK channel activation (Figure 6B). These results indicate that the activation of SK channels could serve as a potential therapeutic approach to normalize AP morphology and reduce the risk of proarrhythmic alternans under pathophysiological conditions with excessive APD prolongation.

### 3.4. Limitations and Future Perspectives

Here, we report that the activation of SK channels in rabbit atria results in APD shortening and that SK channels may serve as a potential antiarrhythmic target. However, our study has some limitations and raises new questions that would need to be addressed in future studies.

The pacing frequency threshold for alternans induction is decreased at lower temperatures [74]. Consequently, at room temperature (20–24 °C), alternans can be studied in isolated myocytes with pacing protocols that are significantly less stressful to the cells, thus extending cell viability over time and allowing for longer stimulation protocols. Furthermore, the kinetic limitations of fluorescent Ca^2+^ indicator dyes make reliable recordings of CaTs at high frequencies difficult at higher temperatures. Specifically, in the case of CaT alternans, the small-amplitude CaT typically starts to fuse with the declining phase of the preceding large-amplitude transient and thus becomes difficult to detect and quantify. Ca^2+^ handling properties and ion channel kinetics are temperature-dependent and are typically faster at physiological temperatures. Thus, AP morphology during alternans might be affected differently at higher temperatures. Nonetheless, previous studies found that electrophysiological properties of the myocardium that are potentially relevant for the development of alternans are not significantly different whether experiments were performed at 30° or 37 °C [75], consistent with a study that found no mechanistic differences between alternans at 27° and 37 °C [76]. SK channel blockers apamin and UCL1684 were shown to inhibit cardiac SK channels at both room [61] and physiological [9,61,77] temperatures. Furthermore, while the function of SK channels could potentially be affected by lower temperatures, several studies, in agreement with our observations, have demonstrated a lack of SK channel contribution to AP repolarization at physiological temperature in rabbit atrial myocytes [12] as well as in other species [12,13,77].

In our study, APs were measured in the whole-cell ruptured-patch configuration. With the ruptured-patch approach, exact control of intracellular [Ca^2+^] is difficult, which could potentially affect SK channel activation. Nevertheless, in rabbit atrial myocytes whole-cell SK current activity under basal conditions could not be demonstrated over a fairly broad range of free [Ca^2+^]_i_ (~100–450 nM) [12]. In addition, studies that used the perforated patch technique, which retains the cellular control of [Ca^2+^]_i_, reported conflicting results, either showing the absence [13,77] or presence [1] of effects of SK channel blockers on cell currents or AP repolarization in different species. An additional layer of complexity of SK channel regulation is given by the fact that in the intact organism, SK channels are under the control of the autonomic nervous system, and sympathetic simulation has been suggested to enhance the recruitment and activity of SK channels [78]. Furthermore, ventricular SK channels appear to exhibit a degree of sexual dimorphism with higher SK channel expression in females [7]. Therefore, it remains to be established how the effects of SK channel activation on AP morphology and cardiac alternans depend on gender and autonomic control, aspects that we did not address in our study.

## 4. Materials and Methods

### 4.1. Ethical Approval

All aspects of animal husbandry, animal handling, anesthesia, surgery, and euthanasia were fully approved by the Institutional Animal Care and Use Committee of Rush University Chicago (IACUC protocol 22-004, approved 25 February 2022; IACUC protocol 22-027; approved 9 May 2022) and comply with the National Institutes of Health’s Guide for the Care and Use of Laboratory Animals. The animals were housed in the dedicated animal housing facility for at least two days before experimentation and had continuous access to drinking water and food.

### 4.2. Myocyte Isolation

Atrial myocytes were isolated from male New Zealand White rabbits (2.5–3 kg; 48 rabbits; Envigo, Indianapolis, IN, USA, and Charles River, Wilmington, MA, USA). Rabbits were euthanized by exsanguination (cardiac excision) under anesthesia. Anesthesia was induced with an intravenous (IV) injection of sodium pentobarbital (100 mg/kg, Euthasol, Virbac AH, Fort Worth, TX, USA) or by isoflurane inhalation. When isoflurane anesthesia was used, 30 min prior to the procedure, the sedative acepromazine (1 mg/kg, Boehringer Ingelheim Animal Health, Duluth, GA, USA) was subcutaneously administered. Anesthesia was initiated with a slow bolus IV injection of propofol (10 mg/kg, Fresenius Kabi, Lake Zurich, IL, USA). Isoflurane (2.5–5%; Piramal Pharma, Digwal, Telangana, India) was applied first through an inhalation mask, and subsequently via a ‘v-gel’ supraglottic airway device (Docsinnovent, Hemel Hempstead, UK) using an isoflurane vaporizer (Tec3; Vetamac, Rossville, IN, USA). The physiological status of rabbits was assessed by blood oxygen saturation and heart rate (Vetcorder Pro monitor; Sentier, Waukesha, WI, USA). To determine the sufficient depth of anesthesia required for intubation, jaw muscle tone, protraction of the tongue, or cough in response to the insertion of the v-gel device were monitored. About 10 min prior to the excision of the heart, rabbits were intravenously injected with heparin (1000 UI/kg; Fresenius Kabi, Lake Zurich, IL, USA). Before the surgical procedure of thoracotomy, the adequate depth of anesthesia was confirmed by foot pinch or checking corneal reflexes. Single atrial myocytes were isolated following a cell isolation protocol that is well-established in our laboratory. As we described previously [50,51,54,64,65], hearts were rapidly excised, mounted on a Langendorff apparatus, and retrogradely perfused via the aorta. Initially heart was perfused for 5–10 min with oxygenated Ca^2+^-free Tyrode solution (in mM): 135 NaCl, 4 KCl, 10 D-Glucose, 5 HEPES, 5 Na-HEPES, 1 MgCl_2_, 1000 UI/l Heparin; pH 7.4 adjusted with 1N HCl. All chemicals and reagents were from MilliporeSigma (St. Louis, MO, USA), unless stated otherwise. Then, the heart was perfused with minimal essential medium Eagle (MEM) (Joklik’s modification, MilliporeSigma, product # M0518) solution containing 20 µM Ca^2+^ and 22.5 µg/mL Liberase TH (Roche Diagnostic Corporation, Indianapolis, IN, USA) for ~20 min at 37 °C. The MEM solution was supplemented with 2 mM Na pyruvate, 10 mM taurine, 10 mM HEPES, 10 mM Na-HEPES, 23.8 mM NaHCO_3_, 50,000 U/L of penicillin, 50 mg/L streptomycin and 40 U/L insulin; pH 7.4 adjusted with 1N HCl. The left atrium was dissected from the heart and minced, filtered, and washed in MEM solution containing 50 µM Ca^2+^ and 10 mg/mL bovine serum albumin. Isolated atrial myocytes were washed and kept in MEM solution with 50 µM Ca^2+^ at room temperature (20–24 °C). Experiments were performed within 1–8 h after isolation.

### 4.3. Patch Clamp Experiments

APs were recorded from single atrial myocytes in the whole-cell ruptured-patch clamp configuration using an Axopatch 200A patch clamp amplifier, the Axon Digidata 1440A interface, and pCLAMP 10.7 software (Molecular Devices, Sunnyvale, CA, USA) as described previously [50,51,54,64,65]. Patch clamp pipettes (1.5–3 MΩ filled with internal solution) were pulled from borosilicate glass capillaries (WPI, Sarasota, FL, USA) with a horizontal puller (model P-97; Sutter Instruments, Novato, CA, USA). The composition of the internal solution was (in mM) 130 K^+^ glutamate, 10 NaCl, 10 KCl, 0.33 MgCl_2_, 4 MgATP, and 10 HEPES with pH adjusted to 7.2 with 1N KOH. The solution was filtered through 0.22-μm pore filters before use. The external Tyrode solution contained (in mM) 135 NaCl, 4 KCl, 2 CaCl_2_, 1 MgCl_2_, 10 HEPES, 10 D-glucose; pH 7.4 with 1 N NaOH. AP recordings were low-pass filtered at 5 kHz and sampled at 10 kHz. For AP measurements, the whole-cell ‘fast’ current-clamp mode of the Axopatch 200A was used, and APs were triggered by 4 ms stimulation pulses with a magnitude ~1.5 times exceeding the threshold of AP activation. Membrane potential (V_m_) measurements were corrected for a junction potential error of −10 mV. APD alternans were induced by increasing the stimulation frequency until stable alternans were observed. The degree of APD alternans was quantified by the ratios of APD_70_ and APD_90_ of pairs of alternating APs. APs were recorded at room temperature (20–24 °C).

### 4.4. Ca^2+^ Measurements

[Ca^2+^]_i_ was measured in atrial cells as we have described previously [51,54]. To increase the adhesion of the cells, glass coverslips were coated with 1 mg/mL laminin. CaTs were triggered by electrical field stimulation with a pair of platinum electrodes with the electrical stimulus set to a voltage ~50% higher than the CaT induction threshold. During the course of experiments, cells were continuously superfused with Tyrode solution. [Ca^2+^]_i_ was monitored using the Ca^2+^ sensitive dye Cal520/AM (AAT Bioquest, Sunnyvale, CA, USA). Atrial myocytes were loaded with 5 μM Cal520/AM for 20–30 min at room temperature and then washed twice for 10 min in Tyrode solution to allow for de-esterification of the dye. Cal520 fluorescence was excited at 485 nm with an Xe arc lamp, and emission fluorescence was recorded at 515 nm using a photomultiplier tube. Background-subtracted fluorescence emission signals (F) were normalized to resting fluorescence (F_0_) recorded under steady-state conditions at the beginning of an experiment. Fluorescence signals were low-pass filtered at 30 Hz. The amplitude of a CaT was calculated as the difference in F/F_0_ (ΔF/F_0_) measured immediately before the stimulation pulse and at the peak of the CaT. Data recording and digitization were achieved using the Axon Digidata 1440A interface and pCLAMP 10.7 software.

To assess SR [Ca^2+^] ([Ca^2+^]_SR_), SR Ca^2+^ release was induced by rapid application of caffeine (10 mM), as we described previously [54]. [Ca^2+^]_SR_ was quantified by the amplitude of the caffeine-induced cytosolic CaT. To ensure consistent SR Ca^2+^ loading, atrial myocytes were paced at 0.5 Hz for at least 2 min before adding caffeine. For paired measurements after the first caffeine-induced CaT, the cell was superfused with control or NS309-containing solution for at least 5 min before the second caffeine application. In half of the cells tested, caffeine-induced CaTs were first measured in control, and subsequently in NS309, and in the other half of cells, caffeine-induced CaTs were first elicited in NS309 treated cells and then recorded after washout of NS309.

### 4.5. CaT Alternans

We used the same approach to induce and quantify CaT alternans as we have described previously [50,51,54,64,65]. Briefly, CaT alternans were induced by incrementally increasing the pacing frequency until stable alternans were observed (the typical range where stable CaT was observed was 1.6–2.5 Hz). The degree of CaT alternans was quantified as the alternans ratio (AR): AR = 1 − [Ca^2+^]_i,Small_/[Ca^2+^]_i,Large_, where [Ca^2+^]_i,Large_ and [Ca^2+^]_i,Small_ are the amplitudes of the large and small CaTs of a pair of alternating CaTs. By this definition, AR values fall between 0 and 1, where AR = 0 indicates no CaT alternans and AR = 1 indicates a situation where SR Ca^2+^ release is completely abolished on every other beat. CaTs were considered alternating when the beat-to-beat difference in CaT amplitude exceeded 10% (AR > 0.1). Once initiated, stable alternans lasted for prolonged periods of time, enabling us to conduct paired experiments. In control conditions, alternans were recorded for at least 1 min before the application of any drug.

### 4.6. Drugs

10 mM NS309 (Cayman Chemical, Ann Harbor, MI, USA), 5 mM UCL1684 (Tocris/Bio-Techne, Minneapolis, MN, USA), and 5 mM HMR1556 (MilliporeSigma, St. Louis, MO, USA) stock solutions were prepared in DMSO. A 1 mM stock solution of apamin (Alomone Labs, Jerusalem, Israel) was prepared in deionized water. Stock solutions were diluted to the final concentration in external solutions. Corresponding amounts of DMSO were added to the control solution (final DMSO concentration ≤ 0.06%).

### 4.7. Data Analysis and Presentation

Results are presented as individual observations or as mean ± SD; n represents the number of individual cells, and N is the number of animals. Where possible, data were collected in a paired manner. Statistical significance was evaluated using one-way ANOVA with Tukey’s test for multiple group comparisons and Tukey’s mixed-effects multiple group comparison test if n within the groups was different. The comparison between the two groups was conducted using a paired Student’s *t*-test for paired data and a Welsh *t*-test for unpaired results. Statistical analysis was performed with GraphPad Prism 9.0 (San Diego, CA, USA). Differences were considered significant at *p* < 0.05. No animals or data points were excluded from the data analysis.

## 5. Conclusions

In this study, we investigated the effect of SK channel modulators on AP morphology and the development of CaT and APD alternans in rabbit atrial myocytes. Our key findings are the following: (i) under basal conditions, block of SK channels does not affect atrial repolarization; (ii) functional SK channels, however, are expressed in the atrial membrane and can be pharmacologically activated, resulting in considerable APD shortening; (iii) activation of SK channels suppressed pacing-induced CaT and APD alternans; (iv) activation of SK channels reduced the amplitude of intracellular Ca^2+^ release and SR Ca^2+^ load; (v) blocking K_V_7.1 channels, mimicking conditions observed in LQT syndrome type-1, increased the risk of atrial CaT alternans which could be abolished by the activation of SK channels. Taken together, our data support the notion that pharmacological modulation of SK channels has the potential to reduce atrial arrhythmia risk resulting from pathological APD changes.

## Figures and Tables

**Figure 1 ijms-26-03597-f001:**
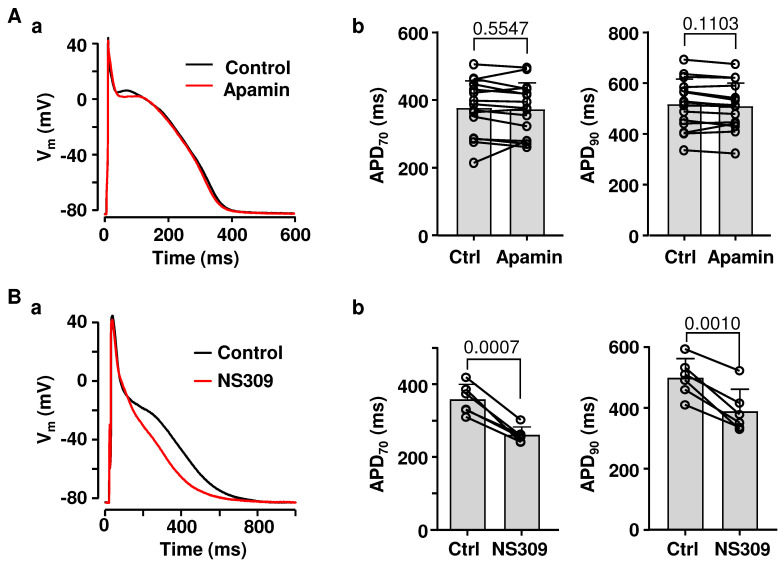
Effect of SK channel inhibition and activation on atrial APD. (**A**): (**a**) Overlay of APs recorded from the same atrial myocyte in control and after application of SK channel blocker apamin (100 nM) at 1 Hz stimulation. (**b**) Mean ± SD and individual APD_70_ and APD_90_ values in control (Ctrl) and in apamin (n/N = 15/10; paired *t*-test). (**B**): (**a**) Overlay of APs recorded from the same cell in control and after application of SK channel activator NS309 (2 µM). (**b**) Mean ± SD and individual APD_70_ and APD_90_ measurements in control and after application of NS309 (n/N = 6/2; paired *t*-test).

**Figure 2 ijms-26-03597-f002:**
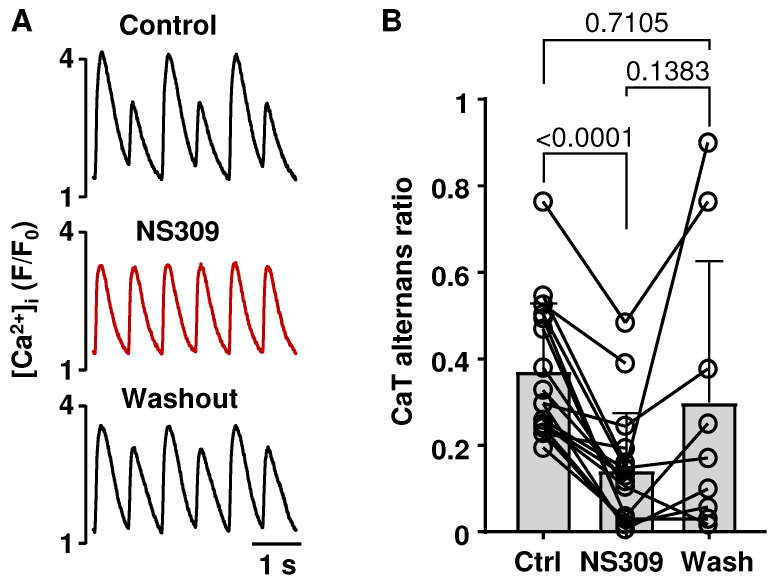
Effect of SK channel activation on CaT alternans. (**A**): CaTs recorded from the same atrial myocyte in control, in the presence of SK channel activator NS309 (2 µM) and after washout of NS309 (stimulated at 1.6 Hz). CaT alternans were induced by increasing field stimulation frequency. (**B**): Mean ± SD and individual cell CaT AR values in control during NS309 exposure (n/N = 17/6) and after washout in a subset of cells (n/N = 9/3). Statistical analysis was performed using Tukey’s mixed-effects groups comparison test.

**Figure 3 ijms-26-03597-f003:**
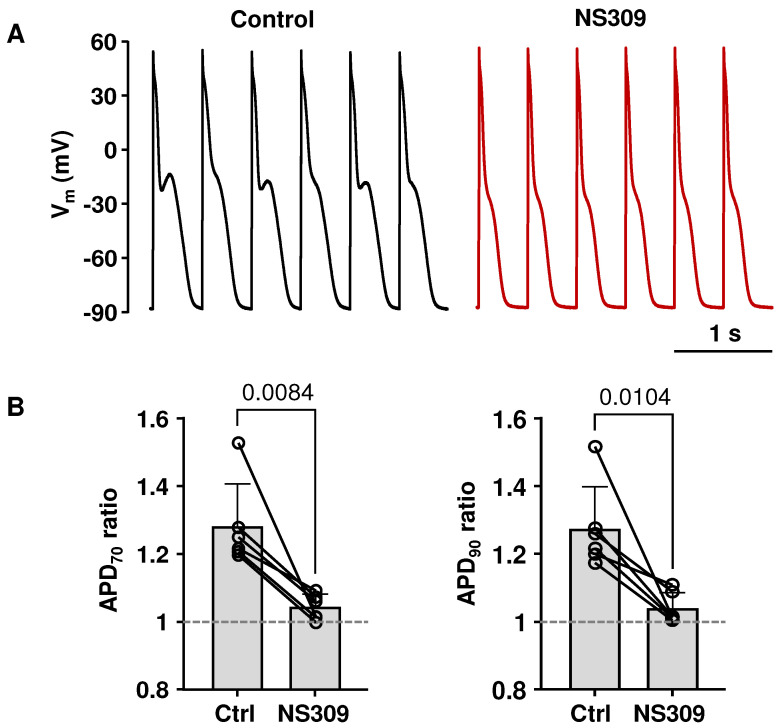
Activation of SK channels abolishes APD alternans. (**A**): APs recorded from the same current-clamped atrial myocyte stimulated at 2 Hz in control and in the presence of SK channel activator NS309 (2 µM). (**B**): Mean ± SD and individual APD_70_ and APD_90_ ratios of pairs of alternating APs in control and in the presence of NS309 (n/N = 6/4). A dashed line marks a ratio of 1, indicating the absence of alternans. Statistical analysis was performed by paired *t*-test.

**Figure 4 ijms-26-03597-f004:**
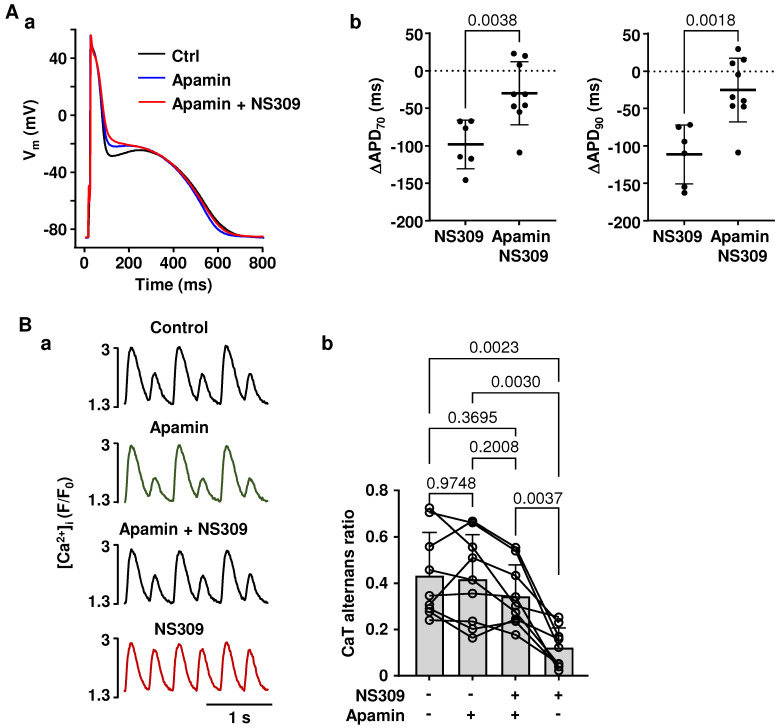
The inhibition of SK channels abolishes NS309’s effects on APD and CaT alternans. (**A**): (**a**) Overlay of APs recorded at 1 Hz from the same atrial myocyte in control, in apamin (100 nM), and in apamin together with NS309 (2 µM). (**b**) Changes in APD_70_ and APD_90_ compared to control (dashed lines) in the presence of NS309 alone (n/N = 15/10) and in the presence of NS309 + apamin (n/N = 9/5). Mean ± SD and individual cell measurements are shown. Cells were pretreated with apamin for 3 min before measurements. Statistical analysis was performed using the Welsh *t*-test. (**B**): (**a**) CaTs recorded from the same field stimulated (2.8 Hz) atrial myocytes in control, in the presence of apamin, during simultaneous application of NS309 and apamin and in NS309 alone. (**b**) Mean ± SD and individual CaT ARs recorded using the experimental protocol from panel Ba (n/N = 9/5); Tukey’s multiple group comparison test.

**Figure 5 ijms-26-03597-f005:**
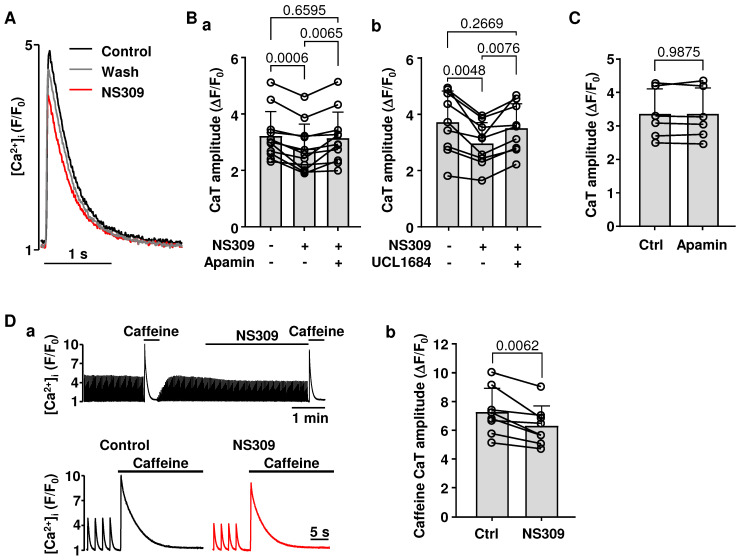
Effect of SK channel activation on CaT amplitude. (**A**): Overlay of CaTs recorded from the same field stimulated (0.5 Hz; no alternans) atrial myocyte in control (black), in NS309 (red), and after washout of NS309 (grey). (**B**): Mean ± SD and individual values of CaT amplitudes (ΔF/F_0_) recorded in control, in the presence of NS309 and during subsequent application of NS309 together with two different SK channel blockers: (**a**) apamin (n/N = 11/3) and (**b**) UCL1684 (1 µM; n/N = 9/3). Statistical analysis was performed using Tukey’s multiple-group comparison test. (**C**): Mean ± SD and individual values of CaT amplitudes observed in control and in the presence of 100 nM of apamin (n/N = 6/2), demonstrating that blocking SK channels had no effect on CaT amplitude. (**D**): (**a**) Caffeine (10 mM) induced CaTs recorded in the same atrial cell (stimulated at 0.5 Hz) before and after NS309 application, shown at different time scales. The top trace shows the decrease in CaT amplitude over time in the presence of NS309. (**b**) Mean ± SD and individual values of caffeine-induced CaT amplitudes (ΔF/F_0_) recorded in control and in the presence of NS309 (n/N = 8/3; paired *t*-test).

**Figure 6 ijms-26-03597-f006:**
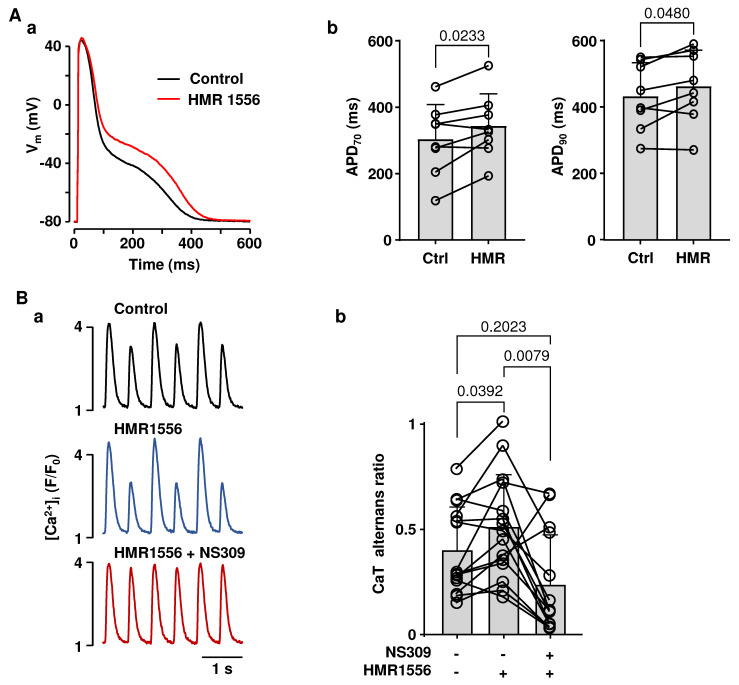
Activation of SK channels reduces the risk for CaT alternans in atrial myocytes with simulated LQT syndrome type-1. (**A**): (**a**) Overlay of APs recorded at 1 Hz from the same atrial myocyte in control and after application of Kv7.1 potassium channel blocker HMR1556 (1 µM) simulating LQT syndrome type-1 conditions. (**b**) Mean ± SD and individual APD_70_ and APD_90_ values recorded in control and in the presence of HMR1556 (n/N = 8/5; paired *t*-test). (**B**): (**a**) CaTs recorded from the same field stimulated (1.8 Hz) atrial myocyte in control, in HMR1556, and during simultaneous application of HMR1556 and NS309. (**b**) Mean ± SD and individual CaT AR values recorded in control, in the presence of HMR1556 (n/N = 14/8), followed by simultaneous application of HMR1556 and NS309 (n/N = 13/7). Statistical analysis with Tukey’s mixed-effects group comparison test.

## Data Availability

The data presented in this study are available on request from the corresponding author.

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
