# Peer review of "Activation of Small Conductance Ca2+-Activated K+ Channels Suppresses Electrical and Calcium Alternans in Atrial Myocytes"

_ijms, 2025, doi:10.3390/ijms26083597_

Round 1
Reviewer 1 Report
Comments and Suggestions for Authors
Small conductance Ca2+-activated K+ (SK) channels are unique in their Ca2+-dependent gating for integrating the intracellular Ca2+ signaling with the membrane potentials. In fact, SK channels play critical roles in central nervous system by their involvement in after-hyperpolarization that follows action potential. In the heart, SK channels were identified to have chamber-specific expression, and participate in the repolarization of cardiac membrane potentials. Due to the essential role of Ca2+ signaling in the regulation of cardiac excitability and contractility, SK channels were proposed to be key ion channels coupling the intracellular Ca2+ with cardiac membrane potentials. However, nearly two decades of study in cardiac SK channels still has not clearly addressed the physiological and pathological mechanisms of cardiac SK channels. In fact, there exist controversial issues in the field regarding the roles of SK channels in the heart.
The authors of this manuscript aimed to address the contributions of SK channels to atrial action potentials (AP), intracellular Ca2+ transients (CaT) and sarcoplasmic reticulum Ca2+ load using rabbit atrial myocytes, and SK channel modulators including inhibitors and activators. They further tested the effects of SK channel activators on pacing-induced AP and CaT alternans. Interestingly, they used a cellular long QT model and tested the effects of SK channel activators on the CaT alternans resulting from long QT. They concluded that SK channel activation in atrial myocytes can shorten atrial AP duration, suppress AP and CaT alternans, and reduce the arrhythmia risks. The question proposed in the study is interesting and important. Investigating the effect of SK channel modulation in CaT and SR Ca2+ load is essential for understanding the feedback mechanisms of SK regulations. The design of the experiment has a plausible rationale, and the results are interesting. I have few comments:
- It is challenging to understand the data shown in Figure 1. Apamin has no effects on APD, but NS309 significantly shortens the APD. Apamin is a specific SK channel inhibitor, and 100 nM apamin is high enough to inhibit all subtypes of SK channels in atrial myocytes if the channels are functionally expressed in the sarcolemma. Did the author think that NS309 will recruit the SK channels to the cell membrane or just activates SK channels in the cell membrane? The immunostaining and membrane protein assay may help to address the expression of SK channels in rabbit atrial myocytes.
- The SK current recording in rabbit atrial myocytes is lacking. The current recording may further address the functional expression of SK channels.
- The temperature for action potential recording was critical. All the recordings of AP were performed at room temperature (20oC-24oC) in the manuscript. In fact, the AP recorded at room temperature is quite different from that at physiological temperature (36-37oC). Physiological condition will provide more insights into the significance of SK channels. In addition, the AP recording was performed by whole-cell recording technique. However, the intracellular Ca2+ concentration in the pipette solution was not quantified and well controlled. This raises a question of how SK channels were activated during AP recording. Will Ca2+ influx provide the Ca2+ source for SK channel activation? This needs to be clarified.
- For apamin+NS309 experiment shown in Figure 4, NS309 has no effects on APD on top of apamin application. This data may not be easily interpreted. The combined effects of apamin and NS309 may depend on the order of the application of apamin and NS309. That says if you use apamin first, then apamin+NS309, or if you use NS309 first, then apamin+NS309. I guess the results may be different because of the different modulatory mechanisms. Please make it clear.
- The weakness of the experiment design is that the authors used only one activator or inhibitor, and did not test the other activators. For increasing the impact of the research, it is better to test different modulators.
- The manuscript was well written. Only concern is that the abbreviated terms need to be accompanied by the full name before they were used in the manuscript. For example, sarcoplasmic reticulum (SR).
Author Response
We thank the reviewer for the time and effort to review our work and valuable comments that help to improving our manuscript. Please, see the attachment for our reply.

Reviewer 2 Report
Comments and Suggestions for Authors
This is an interesting study from authors who have published much over the past decade that has advanced our understanding of the cellular mechanisms underlying cardiac alternans. Cardiac alternans, the beat-to-beat alternation in action potential duration, Ca2+ transient amplitude and/or contractile amplitude, is implicated in the onset of both atrial and ventricular tachyarrhythmias (e.g. atrial fibrillation and torsades de pointes, respectively). In this study, the authors have examined the potential for targeting of Ca2+-activated SK potassium channels to reduce susceptibility to alternans in atrial myocytes. The study follows a previous study from the same authors on targeting SK channels to prevent alternans in rabbit ventricular myocytes (Kanaporis G & Blatter LA. (2023). J Physiol (Lond) 601, 51-67. doi: 10.1113/JP283870)
The study uses atrial myocytes isolated from normal (i.e. healthy/without pathology) male New Zealand white rabbits. Membrane potentials were recorded from single atrial myocytes using whole cell patch clamp, and intracellular [Ca2+] transients (CaT) were measured in separate cells by epifluorescence recording from cells loaded with the Ca2+ fluorophore, Cal520 (485 nm/515 nm) and subject to field stimulation. Experiments were conducted at room temperature. Alternans was induced by progressive incremental reduction in cycle length until stable alternans was induced, typically at 0.4 – 0.66 s CL. Action potential duration (APD) alternans was quantified either as the ratio of a long to a short AP. Ca alternans was quantified as the alternans ratio (AR), calculated as 1 – (small CaT amplitude/large CaT amplitude). During steady state pacing at a CL of 2 s (0.5 Hz), the SK channel blocking toxin, apamin, had no effect on APD at 70 and 90% repolarization (APD70 & APD90, respectively), although the SK channel activator, 2 µM NS309, shortened APD, demonstrating the existence of SK channels as a target for therapeutic intervention. In field-stimulated cells, NS309 reduced the AR and in whole cell recordings, abolished AP alternans (i.e. reduced APD ratio). The effect of NS309 on APD (presumably during steady state pacing) was inhibited by apamin. The effect of NS309 on CaT AR was abolished in the presence of apamin. NS309 reduced the amplitude of CaT during steady state pacing, and this was associated with a reduction in the amplitude of the caffeine-induced CaT (an index of the Ca2+ content of the sarcoplasmic reticulum). The Kv7.1/IKs channel blocker, HMR1556, produced APD prolongation during steady state stimulation and increased CaT AR in field-stimulated cells. The effect of HMR1556 on CaT AR was largely abolished by NS309. The authors conclude that the pharmacological modulation of SK channels has the potential to reduce atrial arrhythmia risk from pathological changes in APD.
Specific comments
- Discussion, lines 273 – 280. “Our results indicate that under basal conditions in rabbit hearts inhibition of SK channels has no effect on APD duration in atrial myocytes.” This statement assumes that all SK channels are apamin-sensitive. Published evidence indicates that this may not be the case. Heteromultimeric SK2-SK3 channels are relatively apamin-insensitive (Hancock et al. (2015). Heart Rhythm 12, 1003-1015. doi: 10.1016/j.hrthm.2015.01.027). On the other hand, SK2-SK3 channels and mouse atrial myocyte APD were shown to be sensitive to the alternative SK channel blocker, UCL1684. Evidence was also presented that UCL1684 was selective for SK channels over other cardiac channel types. Have the authors tested the effects of this blocker on rabbit atrial APD and alternans? In any case, this section of the discussion should be re-written, acknowledging the possibility that not all SK channels may be sensitive to apamin, citing the Hancock et al study.
- The authors should include some comment on the significance of conducting the experiments at room temperature on the observation of CaT and AP alternans. The rate of ATP production and the transport rate via the SR Ca2+ ATPase can be expected to be highly sensitive to experimental temperature (e.g. Bers DM & Bridge JH. (1989). Circ Res 65, 334-342. doi: 10.1161/01.RES.65.2.334). The cycle lengths at which alternans were observed are within the usual range of resting cycle lengths in the rabbit heart. With this in mind, what is the pathological significance of the mechanisms observed in this study?
- Provide a quantitative definition of ‘stable alternans’. Were exclusion criteria used to define stable alternans (e.g. minimum AR ratio, minimum duration for an episode of ‘stable alternans’)? If so, what were these? Was the duration of episodes of alternans changed in the presence of NS309 relative to control? Was there a difference between control and NS309 in the CL at which stable alternans was induced? What was the effect of HMR1556 on the duration of episodes of alternans? Was there a difference between control and HMR1556 in the CL at which stable alternans was induced? If the CL was shortened further in the presence of NS309, would the AR increase? These details should be provided to the reader.
- Provide the stimulation frequency/cycle length in the figure legend for the recordings of AP in the steady state (i.e. Figure 1, Figure 3, Figure 4 and Figure 6).
Author Response

(The authors gave the same response as above.)

Round 2
Reviewer 1 Report
Comments and Suggestions for Authors
The authors addressed some of my previous questions. However, there are still concerns regarding several questions.
- As shown in Figure 1, apamin has no effects on APD, but NS309 significantly shortens the APD. The authors stated that “SK channels are functionally expressed and present in the sarcolemma of atrial myocytes, even though under our experimental conditions SK channels are not activated at basal conditions.” I wonder what the baseline conditions are. If the channels are expressed, and if the intracellular Ca2+ concentration is kept high enough to activate the SK channels during action potentials, the SK channel should be activated following the Ca2+ dynamics. This is a fundamental question of this study. More efforts and time are needed to address this.
- The SK current recording in rabbit atrial myocytes is lacking. The current recording may further address the functional expression of SK channels. If the authors think NS309 can activate SK channels, the current recording will provide the experimental evidence.
- The central parameter of this study is AP, and AP is very sensitive to temperature. I still think the AP recording at physiological condition is necessary at least for testing the apamin and NS309 effects. Proper control of intracellular Ca2+ concentration is essential for SK current recordings.
Author Response
Thank you for providing the review for resubmitted manuscript ijms-3505004.
We would appreciate your advice. We strongly believe that the reviewer's requests for revision are far beyond the scope of our study and by no means could possibly be achieved in ten days. The reviewer requests additional experimental work that amounts essentially to a new paper:
- concern 1: to address this concern the reviewer essentially requests to repeat all experiments with simultaneous action potential AND intracellular [Ca]i measurements. These simultaneous Vm and [Ca]i measurements are demanding experiments and constitute an entirely new and different study. We decided in our current study to characterize the involvement (or lack thereof) of SK channels using pharmacological tools. We are using activators and blockers of SK channels that are generally considered to be specific and effective.
- concern 2: Above argument applies largely also to this concern. To provide electrophysiological SK current measurements in addition to the pharmacological results that we present in our study, in our opinion, don't provide any additional insights, particularly given the fact that the activator NS309 is considered to be specific, and our use of 2 different SK channel blockers that prevent or reverse the effects of NS309 constitutes strong evidence that NS309 indeed activates SK channels and our observed effects on cellular alternans are indeed mediated by SK channels.
- concern 3: The reviewer essentially requests to repeat the study at body temperature. This is an endeavor that is way beyond the scope of our study. We believe we have given a detailed explanation in the revised manuscript why we conducted our study at room temperature and we have discussed critically the related limitations.
We would highly appreciate your editorial opinion and advice on the scope of the requested revisions.
Sincerely,